# Far-field acoustic subwavelength imaging and edge detection based on spatial filtering and wave vector conversion

Chu Ma [1], Seok Kim[1] & Nicholas X. Fang[1]

The resolution of acoustic imaging suffers from diffraction limit due to the loss of evanescent field that carries subwavelength information. Most of the current methods for overcoming the diffraction limit in acoustics still operate in the near-field of the object. Here we demonstrate the design and experimental realization of an acoustic far-field subwavelength imaging system. Our system is based on wave vector filtering and conversion with a transmitter at the near-field and a spatially symmetrical receiver at the far-field. By tuning geometric parameters of the transmitting/receiving pair, different spatial frequency bands can be separated and projected to the far-field. Furthermore, far-field imaging and edge detection of subwavelength objects are experimentally demonstrated. The proposed system brings new possibilities for far-field subwavelength wave manipulation, which can be further applied to medical imaging, nondestructive testing, and acoustic communication.

---

[1] Department of Mechanical Engineering, Massachusetts Institute of Technology, 77 Massachusetts Avenue, Cambridge, MA 02139, USA. Correspondence and requests for materials should be addressed to N.X.F. (email: nicfang@mit.edu)

Acoustic imaging techniques have been widely used in areas such as medical ultrasonic imaging[1], nondestructive testing[2], and underwater sonar systems[3,4]. Similar to other wave-based imaging techniques, the high spatial frequency information of an object is carried by evanescent waves that decay exponentially as they leave the object. The loss of those evanescent waves in conventional imaging systems leads to the diffraction limit[5]. The working wavelength of acoustic imaging is relatively large compared to imaging techniques using other wave forms (e.g., optical wave, x-ray, electron beams, microwaves, etc). Thus, the resolution upper bound posed by the diffraction limit has larger impact on acoustic imaging. Various approaches have been developed to realize acoustic imaging beyond the diffraction limit[6–20].

The near-field scanning technique[6] is one of the earliest techniques for acoustic subwavelength imaging. It requires measurements to be made in close proximity (fractions of wavelength) to objects in order to capture the evanescent components. Such restriction greatly limits its application in acoustic subwavelength imaging. In the past decade, acoustic metamaterial-based lenses have demonstrated the potential in subwavelength imaging[10–15]. The dispersion relation and effective material properties were modified inside the metamaterials to support the propagation of evanescent waves. By coupling to acoustic metamaterial lenses in the near-field of the objects, the evanescent waves can be measured further away. Although the distance is extended by using existing acoustic metamaterials, the thermal viscous loss due to wave propagation in resonating elements is still a major factor that limits the use of metamaterial-based lenses in subwavelength imaging.

If the evanescent wave can be converted to propagating wave and propagate in free-space or empty waveguide, the propagation distance will be much less limited by thermal viscous loss and the image can be formed in the far-field of the object. One of the techniques for far-field acoustic imaging is hyperlenses[19,20], which use cylindrical-shaped anisotropic metamaterials to couple and gradually compress large subwavelength wave vectors to small propagating wave vectors. At the exit of a hyperlens, the image size is larger than the wavelength and the image can be detected in the far-field. However, due to the geometric shape, achieving higher spatial resolution of hyperlenses has a cost of smaller field of view. Another technique for acoustic far-field imaging is the time reversal technique[16–18], which uses random subwavelength scattering arrays to convert evanescent waves to propagating waves in the near field, in combination with a time reversal mirror in the far-field. However, time reversal imaging is not a direct projection based technique. It requires a complex recording/playback system and the image needs to be formed at the same location as the object. Those requirements reduce the flexibility for applications. Far-field superlenses were designed in optics to realize far-field subewavelength imaging. They were designed by using gratings to convert optical evanescent waves to propagating waves[21,22]. The converted propagating waves were measured in far-field free space and reconstructed computationally. In order to apply gratings to imaging, the overlapping of multiple diffraction orders needs to be addressed. In the design of optical far-field superlens, the overlapping was removed by combining the grating with a metal layer that can excite subwavelength surface plasmon to couple selected spatial frequencies to the grating. However, no counterpart of surface plasmon exists in acoustics. Thus, no such far-field superlens has been realized in acoustics yet.

In this work, we propose mechanisms in acoustics to realize the functions of optical surface plasma and grating, and combine them to demonstrate an acoustic far-field subwavelength imaging system. A resonator array (denoted as filter layer) of subwavelength unit cells is designed to realize the function of surface

plasma in order to amplify selected subwavelength spatial frequencies. An acoustic version of binary phase grating[23] (denoted as grating layer) is designed to remove the incident propagating components and convert the incident subwavelength components to propagating ones. The combination of those two proposed structures forms the transmitter of the system, and establishes the one-to-one relationship between the subwavelength wave vectors and the propagating ones. Furthermore, instead of measuring the converted propagating wave directly and performing the back conversion in computer as implemented in the optical far-field superlens, we use a receiver that is spatially symmetrical to the transmitter to physically reconvert the propagating wave to the original evanescent wave in the far-field of the object, resulting in a reciprocal transmitting/receiving pair. In the proposed system, wave propagation takes place in empty waveguide with much less thermal viscous loss than in metamaterials. We show that while the transmitter still needs to be close to the object, the receiver can be many wavelengths away from the object and the distance can be flexible. This is the reason why we call the designed system far-field lenses. Our lenses work as spatial filters that separate different subwavelength spatial frequencies and project them to the far-field. Edge detection is experimentally demonstrated as an example of far-field sub-wavelength imaging of our proposed system.

## Results

**System model.** Fig. 1a illustrates the system configuration that will enable the transmitter/receiver to faithfully convert evanescent wave to propagating wave and vice versa. As a proof of concept, the system is set in air with sound speed $c_{air} = 343$ m/s. The object here is assumed to be acoustic scattering media with subwavelength details and is put in contact with the filter layer in the transmitter. The sound source emits continuous wave at frequency $f_0$ (corresponding to wavenumber $k_0$ in air) or a short pulse with center frequency $f_0$. The scattered sound wave from the object, $P_0 = \int_{-\infty}^{\infty} p_0(k)e^{j(kx-2\pi f_0 t)}dk$, enters the filter layer in the transmitter. $P_0$ is expressed as the integration of the Fourier spectrum for the scattered wave from the object, based on the principle of spatial Fourier transform[24]. $P_0$ contains both propagating waves ($|k| < k_0$) and evanescent waves ($|k| \geq k_0$). The function of the filter layer is to amplify the wave components with a selected wavenumber band. After passing through the filter layer, the filtered wave is now $P_1 = \int_k p_1(k)e^{j(kx-2\pi f_0 t)}dk$, where $|k| \in [k_G - k_0, k_G]$ or $|k| \in [k_G, k_G + k_0]$ (In Fig. 1a we plot the former case.). $k_G = 2/\gamma$ is the grating constant corresponding to grating period $\gamma$. The function of the grating layer is to convert the subwavelength wave vectors emerging from the filter layer to propagating wave vectors by momentum addition. After passing through the grating layer, $P_1$ will be converted to $P_2 = \int_{k'} p_2(k')e^{j(k'x-2\pi f_0 t)}dk'$. If $k > 0$, the conversion is through the $-1$st order diffraction of the grating and $k' = k - k_G$. If $k < 0$, the conversion is through the $+1$st order diffraction of the grating and $k' = k + k_G$. The grating is designed to only have odd diffraction orders, thus the propagating components from the filter layer, which is the zeroth order diffraction, is further suppressed. The receiver is composed of the same grating layer and the filter layer as the transmitter. By reciprocity, the receiver takes the reverse process. The grating layer first adds $k_G$ to or subtracts $k_G$ from $k'$ to obtain $P_3 = \int_k p_3(k)e^{j(kx-2\pi f_0 t)}dk$. The amplitudes $p_1(k)$, $p_2(k')$ and $p_3(k)$ are derived in later sections as transmission coefficients of filter layers and grating layers. The grating will also generate other higher odd order diffractions (3th order, 5th order, …). Then the filter layer in the receiver will perform post-filtering for the waves with wave number $|k| \in [k_G - k_0, k_G]$ or $[k_G, k_G + k_0]$ in the first order diffraction. Other higher-order diffractions

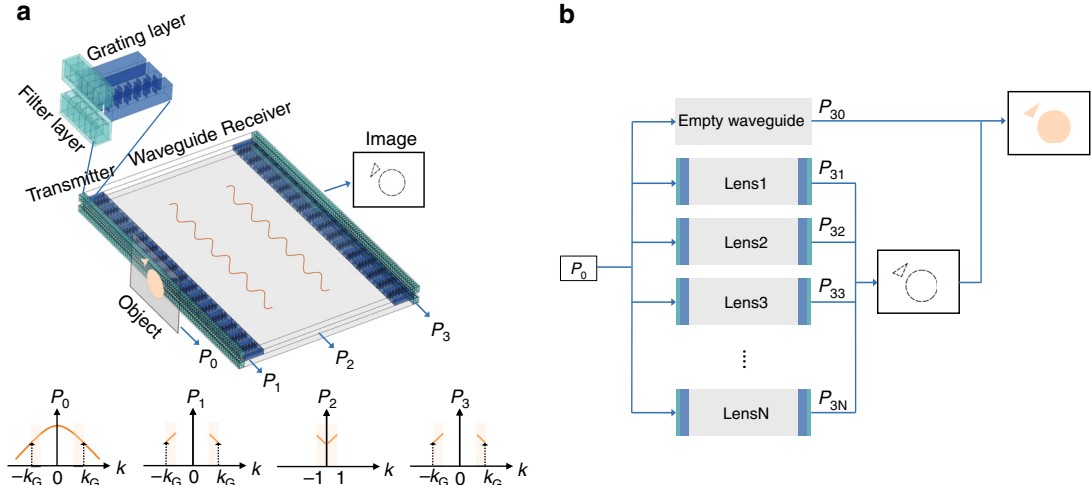

**Fig. 1** System schematics and the working process of the proposed system. **a** System schematics. The whole system is composed of a transmitter and a receiver that are spatially symmetrical. Each part is composed of a filter layer (green) and a grating layer (blue). Evanescent wave $P_1 = \int_k p_1(k)e^{j(kx-2\pi f_0 t)} dk$, where $|k| \in [k_G - k_0, k_G]$, will be filtered out by the filter layer from the scattered wave by the object $P_0 = \int_{-\infty}^{\infty} p_0(k)e^{j(kx-2\pi f_0 t)} dk$, and be converted to propagating wave $P_2 = \int_{k'} p_2(k')e^{j(k'x-2\pi f_0 t)} dk'$, ($k' = k - k_G \in [-k_0, 0]$ when $k > 0$ or $k' = k + k_G \in [0, k_0]$ when $k < 0$) by the grating layer in the transmitter. In the receiver, $P_2$ will be converted back to evanescent wave $P_3 = \int_k p_3(k)e^{j(kx-2\pi f_0 t)} dk$. **b** Imaging process. By tuning the geometric parameters of filter layer and grating layer, different subwavelength spatial frequency bands of the scattered wave from object ($P_0$) can be separately projected to far-field ($P_{3n}$, $n = 1, 2, 3, \ldots$). The propagating band can be obtained by propagating through empty waveguide. The full subwavelength image can be obtained by summing up all the bands $\left(P_I = \sum_{n=0}^{N} P_{3n}\right)$. The edge of the object can be detected by adding up just the subwavelength components $\left(P_E = \sum_{n=1}^{N} P_{3n}\right)$

are all attenuated after passing through the filter layer at the receiver. In order to prevent the overlapping of +1st and −1st diffraction orders of the grating layer, the wavenumber bands of the wave after passing through the filter layer cannot locate in $[k_G - k_0, k_G]$ and $[k_G, k_G + k_0]$ simultaneously. By changing the geometric parameters, the filtering band of the filter layer and the grating constant $k_G$ of the grating layer can be adjusted to different locations in the spatial spectrum. Thus we can separately project the bands $|k| \in [nk_0, (n+1)k_0]$, $n = 1, 2, 3, 4, \ldots N$, to the far-field, as shown in Fig. 1b. The received signals are denoted by $P_{3n}$ respectively. The propagating waves with $|k| \in [0, k_0]$, denoted by $P_{30}$, will be obtained by wave propagating through an empty waveguide without transmitter/receiver. The final image can be formed in two ways. One way is summing up all the components together $\left(P_I = \sum_{n=0}^{N} P_{3n}\right)$. In this case, a full image with subwavelength details $|k| \in [0, (N+1)k_0]$ will be obtained. The other way is summing up only the subwavelength components $\left(P_E = \sum_{n=1}^{N} P_{3n}\right)$. In this case, the edge image of the object that is represented by higher spatial frequencies will be obtained. To demonstrate the concept in this work, we will show three different configurations that work for $|k| \in [k_0, 2k_0]$, $[2k_0, 3k_0]$, and $[3k_0, 4k_0]$.

**Transmission properties of filter layer**. The purpose of filter layer is to couple incident waves with specific wave numbers from input to output of the layer, as a near-field spatial filter. In optics, such filtering is realized through the excitation of surface plasma in a metallic layer. However, there is no counterpart of surface plasma in acoustics yet. We propose to realize the near-field spatial sound filtering by using two identical arrays of Helmholtz resonators (HRs) facing each other with distance $h$, forming a waveguide of height $h$ for the wave to pass through (Fig. 2a). Here $h$ is smaller than the wavelength $\lambda_0$ in free space to guarantee that the wave vector in $y$ direction is zero. Each HR in the arrays is built with a cavity having widths $a_1$ and $c$, and height $b_1$, and a

rigid tube having widths $a_2$ and $c$, and height $b_2$. The array period is $d_1$. The resonance frequency of the HR corresponds to a wavelength that is much larger than the size of the HR itself, so we can use lumped circuit model to approximate the HRs[25], with the cavity acting as the capacitor and the tube acting as the inductor.

When the incident field propagates through the waveguide, some components with specific transverse wave numbers will be coupled to the resonance mode of the HR arrays. The HRs in the array should have much smaller sizes and periods compared to the working wavelength in order to make the coupling of subwavelength components possible. We choose our working frequency as $f_0 = 9000$ Hz, corresponding to wavelength of $\lambda_0 = 38.1$ mm, for demonstration. The proposed system can be scaled to other sizes and working frequencies depending on different applications. In the coordinate system shown in Fig. 2a, the incident field at the entrance of the filter layer has pressure distribution $p_{in} = e^{ik_0 k_e x}$, where $k_e = k/k_0$ is the effective wave vector in the $x$ direction. The output pressure field distribution $p_{out} = p_1 e^{ik_0 k_e x}$ along the $x$ direction is obtained at the exit of the filter layer. The transmission coefficient of the filter layer, $T_f(k_e)$, is defined as $T_f = p_{out}/p_{in}$. It is the coupling of wave component with effective vector $k_e$ from the input to the output of the filter. The theoretical transmission coefficient is calculated using lumped circuit model (see Supplementary Note 1 for details) as follows:

$$T_f(k_e) = \frac{s_0}{d_1^2 k_e^2 k_0^2 + 2YZ} e^{jk_e k_0 x}, \tag{1}$$

where $s_0$ is a constant indicating the strength of the coupling, $Z$ is the effective impedance and $Y$ is the effective admittance. $Y$ and $Z$ are functions of the geometric parameters of the HR arrays. When the denominator $(d_1^2 k_e^2 k_0^2 + 2YZ)$ approaches zero, the resonance occurs and the transmission through the waveguide between the HR arrays is largely enhanced. Equation 1 is derived

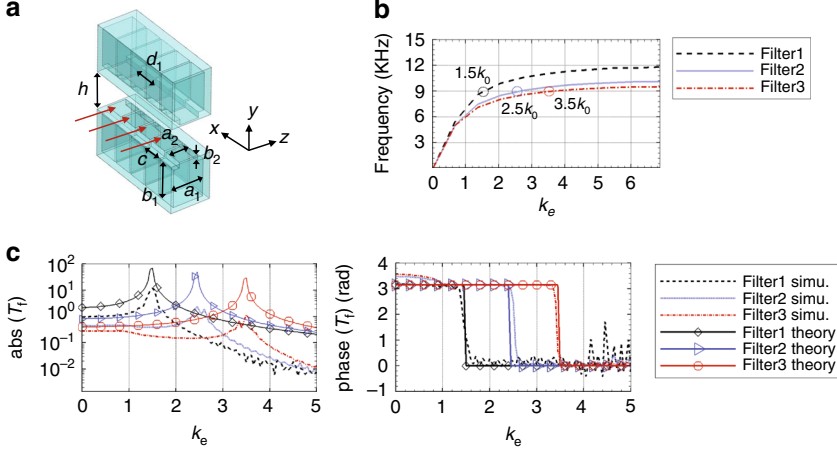

**Fig. 2** Model and properties of the filter layer. **a** Model for filter layer. The geometric parameters are: $b_1 = 5$ mm, $a_2 = 2.5$ mm, $c = 2.5$ mm, $b_2 = 0.75$ mm, $h = 5$ mm, and $d_1 = 2.75$ mm. By changing $a_1$, the filtered wave vector can be tuned. We choose $a_1 = 4.5$ mm for $k_e = 1.5$ (filter1), $a_1 = 6$ mm for $k_e = 2.5$ (filter2), and $a_1 = 6.5$ mm for $k_e = 3.5$ (filter3). **b** Theoretically calculated dispersion relations of filter1, filter2 and filter3 in the x direction. The dispersion relation curves of filter1, filter2 and filter3 pass through $k_e = 1.5$, $k_e = 2.5$ and $k_e = 3.5$, respectively at $f_0 = 9000$ Hz. **c** Amplitudes and phases of theoretically calculated and simulated transmission coefficients of filter layers

based on the assumptions that no damping is considered and the array is infinitely long. This is comparable to eigenmode analysis. In realistic experiments or simulations, the arrays are of finite length and damping has to be considered, so that the transmission will not diverge. From $d_1^2 k_e^2 k_0^2 + 2YZ = 0$, we get $k_e = \sqrt{-\frac{2YZ}{d_1^2 k_0^2}}$, which determines the wave number that is amplified by the filter layer. By changing the cavity size $a_1$ of the HRs, we obtained that $k_e = 1.5$ for the filter layer with $a_1 = 4.5$ mm (denoted by filter1), $k_e = 2.5$ for the filter layer with $a_1 = 6$ mm (denoted by filter2), and $k_e = 3.5$ for the filter layer with $a_1 = 6.5$ mm (denoted by filter3). The theoretical calculation of $s_0$ in Eq. 1 needs multiple scattering theory and mode analysis for the waves inside HRs and the waveguides. Here instead, we use COMSOL full wave simulation to calculate $s_0$. Two HR arrays, each containing 90 HRs, are used in the simulation (see Supplementary Note 4 for detailed description of the COMSOL simulation). By matching the theoretical and simulated $T_f$ at $k_e = 0$, we get $s_0 = 2.35$, 1.85 and 1.5 for three curves respectively.

For each $k_e \in [0, 5]$, the amplitudes and phases of both theoretical and simulated $T_f$ for the three values of $a_1$ are plotted in Fig. 2c. Both theoretical and simulation results show that the amplitude is amplified at $k_e = -\frac{2YZ}{d_1^2 k_0^2}$. Neither case considers loss. The amplification obtained in the simulation is in general smaller than that in the theoretical calculation. It is because that in the simulation the filter layer has finite numbers of HRs, which is unable to produce the resonant amplitude that approaches infinity shown in the theoretical calculation for infinite structures. Both theory and simulation indicate a $\pi$ phase shift before the resonance wave vector and zero phase shift after the resonance wave vector.

The dispersion relations of the filter layers in x direction for three different $a_1$ values are calculated (see Supplementary Note 1) and plotted in Fig. 2b. The dispersion relation curve shifts to lower frequencies as $a_1$ increases, resulting in larger $k_e$ corresponding to $f_0 = 9000$ Hz. By further increasing $a_1$, the filter layer can work for wave vectors larger than $3.5k_0$. The largest wave vector that can be accessed is determined by $k_e \le \frac{\pi}{d_1 k_0}$, where $d_1$ is the period of the HR array. In our case, $d_1 = 2.75$ mm, so $k_e \le 6.9$.

**Transmission properties of grating layer**. In our designed system, a grating layer is put next to the filter layer to convert the subwavelength wave to propagating wave. A grating usually has multiple diffraction orders. The key here is to establish a one-to-one relationship between the wave vectors before and after conversion. Observing the simulated transmission coefficients in Fig. 2b, we find that the propagating wave components ($k_e \in [0, 1]$) are still large compared to the filtered subwavelength components, especially for filter layers of higher spatial frequencies. Those unwanted propagating waves will transmit through the grating layer by the zeroth order diffraction and overlap with the subwavelength components of interest that transmit by the first order diffraction. Binary phase grating in optics is known to not have even order diffraction[23]. Thus we propose the design of an acoustic binary phase grating to eliminate the zeroth order diffraction of the propagating components.

Ideally, the binary phase grating is composed of elements that produce uniform amplitude transmission and alternating phase shifts of 0 and $\pi$. Practically, it is difficult to find materials or structures that have perfect impedance matching with air in order to generate full transmission while also have sound speed variation in order to generate $\pi$ phase shift. Narrow acoustic waveguide channels with cross section dimension smaller than wavelength are known to be able to generate phase delay proportional to the wave path length[26]. For waveguides with wave path length of $\lambda_0/2$ and $\lambda_0$ that generate phase shifts of $\pi$ and $2\pi$, the Fabry-Perot resonances can be excited inside the tubes to achieve high transmission efficiency[27]. Based on those facts, we construct the acoustic binary phase grating in our system with a straight channel of length 17.5 mm (around $\lambda_0/2$) and a curved channel with wave path length of around $\lambda_0$ but size of 17.5 mm in the z direction, as shown in Fig. 3a. The width of the channels is 1.25 mm, which is much smaller than the working wavelength. The height of the channels, $h = 5$ mm in the y direction, is the same as the height of the waveguide between the two HR arrays.

For a binary phase grating with period $2d_2$, the grating constant is $k_G = \pi/d_2$. The transmission coefficient $T_g^t(k_e)$ for the grating is defined as the coupling efficiency from subwavelength components $k_e k_0$ to the propagating components $k_e k_0 - n k_G$ through the $-n$th order diffraction. $T_g^r(k_e)$ is defined as the coupling efficiency from the propagating components $k_e k_0 - n k_G$ to the subwavelength

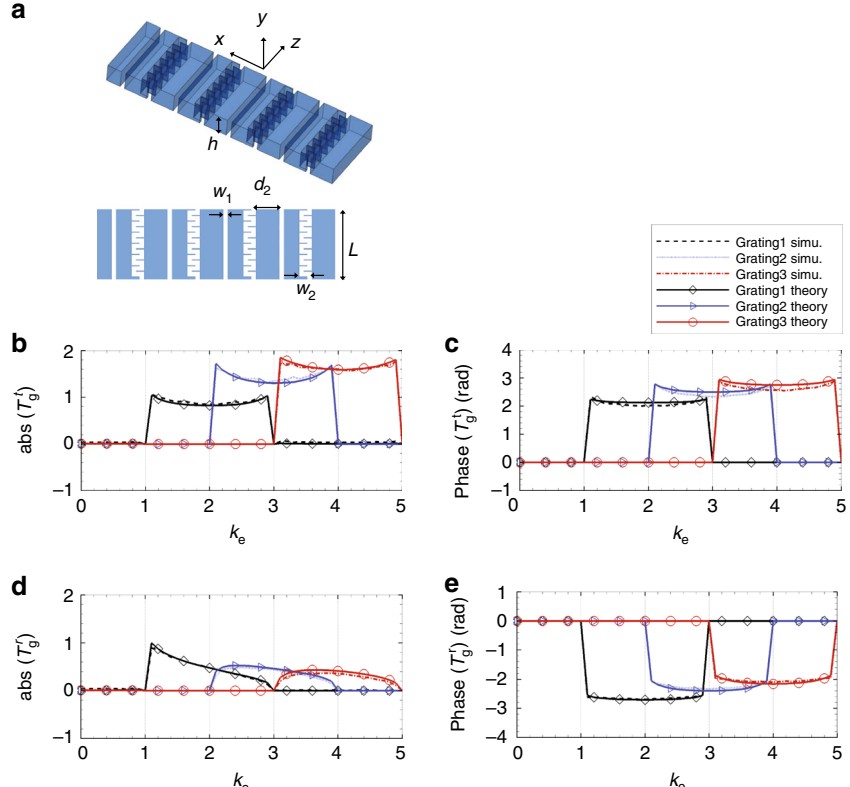

**Fig. 3** Model and properties of the grating layer. **a** Model for grating layer. Geometrical parameters are: $w_1 = 1.25$ mm, $w_2 = 6.95$ mm, $L = 18.75$ mm. Changing the grating period $d_2$ will results in different $k_G$. We choose $d_2 = 18.9$ mm for $k_G = 2k_0$ (grating1), $d_2 = 12.6$ mm for $k_G = 3k_0$ (grating2), and $d_2 = 9.4$ mm for $k_G = 4k_0$ (grating3). **b**, **c** Amplitudes (**b**) and phases (**c**) of theoretical and simulated transmission coefficients of grating layers in the transmitter. **d**, **e** Amplitudes (**d**) and phases (**e**) of theoretical and simulated transmission coefficients of grating layers in the receiver

components $k_e k_0$ through the $+n$th order diffraction ($k_e \in [n-1, n+1]$, $n = 0, 1, 2, \ldots$). $T_g^t(k_e)$ and $T_g^r(k_e)$ are calculated with plane wave expansion method[27] (see Supplementary Note 2 for detailed calculation). The waves inside the channels and at both sides of the grating are expressed as combinations of plane wave modes. The coefficients of different modes are computed by matching pressures and perpendicular velocities at the two interfaces of the grating.

In order to verify the theoretical approach, COMSOL full wave simulation is performed to calculate the $T_g^t(k_e)$ and $T_g^r(k_e)$ of the grating with size of 250 mm in the $x$ direction (see Supplementary Note 4 for detailed description of the COMSOL simulation). Figs. 3b–e show the amplitudes and phases of the theoretical and simulated transmission coefficients $T_g^t(k_e)$ and $T_g^r(k_e)$ of the three phase gratings with different grating periods corresponding to $k_G = 2k_0$, $3k_0$, and $4k_0$ (denoted by grating1, grating2, and grating3, respectively). Both theoretical calculation and simulation show that the grating layer demonstrates large −1st and +1st order diffraction for efficient subwavelength/propagating wave vector conversion as well as near-zero zeroth order diffraction for removing the influence of propagating components in the incoming wave. $T_g^t(k_e)$ gives positive phase shifts and $T_g^r(k_e)$ gives negative phase shifts.

Smaller channel width $w_1$ will decrease the unit cell size in $x$ direction, enabling larger $k_G$ for converting larger subwavelength wave vectors to propagating wave vector range. However, a smaller channel width will generate more thermal viscous loss inside the channels. The choice of $w_1$ here is a trade-off among those factors at frequency 9000 Hz.

**Image transfer function of the whole system**. In the whole system shown in Fig. 1a, the receiver will reconvert the propagating waves converted from subwavelength waves by the

transmitter back to the original forms. Three filter layer/grating layer combinations are considered: (1) $a_1 = 4.8$ mm, $k_G = 2k_0$, (2) $a_1 = 5.9$ mm, $k_G = 3k_0$, and (3) $a_1 = 6.5$ mm, $k_G = 4k_0$ (denoted by lens1, lens2 and lens3, respectively). The distance between the transmitter and the receiver is $D$ in all three combinations. The waves propagate in empty waveguide of length $D$ with negligible loss.

When calculating the transmission coefficient of the whole system (denoted as image transfer function), we need to consider two cases. One case is when continuous wave being used as the sound source. Multiple reflections between the two gratings need to be taken into consideration. In this case, the plane wave expansion method is applied to the combination of the two gratings and the space between them (see Supplementary Note 3 for detailed derivation). The calculated transmission coefficient for the combination of two gratings is denoted as $T_g(k_e)$. The other case is when a short pulse being used as the sound source. Similar to 'single pass' of signals, all multiple reflections are neglected because their arrival time at the receiver does not overlap with the direct arrived signal. In this case, we only need to consider the phase delay in the empty waveguide from transmitter to receiver. The transmission coefficient of the two gratings and the space between is expressed as

$$D_g(k_e) = T_g^t T_g^r e^{-jD\sqrt{k_0^2 - (k_e k_0 - k_G)^2}}, \quad k_e \in \left[\frac{k_G}{k_0} - 1, \frac{k_G}{k_0} + 1\right].$$

The image transfer function can be expressed as $T_{continuous} = T_f T_g T_f$ (when using continuous source) or $T_{pulse} = T_f D_g T_f$ (when using pulse source), indicating the coupling strength of subwavelength wave component with effective wave vector $k_e$ from the input side to the output side. When $D = 130$ mm ($3.4\lambda_0$), the amplitudes of theoretically calculated $T_{continous}$ and

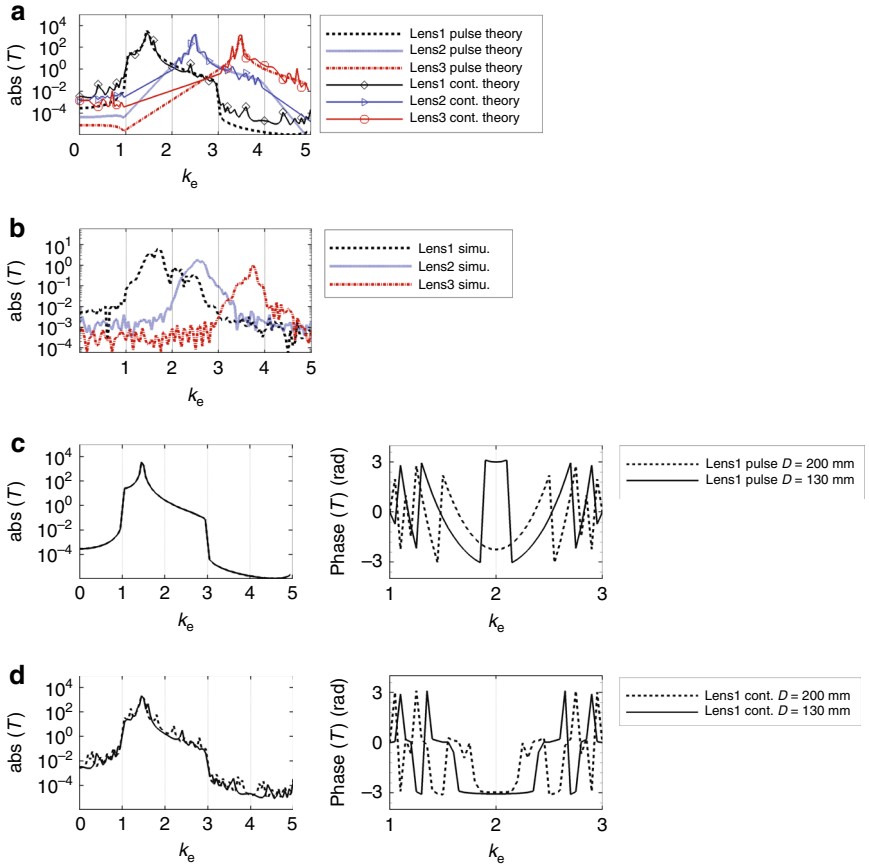

**Fig. 4** Transmission properties of the whole system. **a** Amplitudes of theoretically calculated image transfer functions of lens1, lens2 and lens3 for pulse source case and continuous source case. Lens1, lens2 and lens3 correspond to systems that work for wave vectors in $k_e \in [1, 2]$, $[2, 3]$, and $[3, 4]$, respectively. **b** Amplitudes of simulated image transfer function of lens1, lens2 and lens3. Simulation is performed in frequency domain, taking into consideration the multiple reflections between gratings. **c** When using continuous source, theoretically calculated amplitudes and phases of image transfer functions for lens1 when $D = 130$ mm and $D = 200$ mm. $D$ is the distance between the transmitter and the receiver. **d** When using pulse source, theoretically calculated amplitudes and phases of image transfer functions for lens1 when $D = 130$ mm and $D = 200$ mm

$T_{pulse}$ for lens1, lens2 and lens3 are plotted in Fig. 4a. Since the transmission coefficient of the filter layer has a much larger magnitude than that of the grating layer near resonance, both amplitudes of $T_{continous}$ and $T_{pulse}$ are dominated by the trend of filter layer amplitude response. The image transfer functions of the three lenses are also obtained in COMSOL frequency domain simulation. Fig. 4b shows the amplitudes of the simulated image transfer functions. Figs. 4a, b both indicate that the subwavelength waves with wavenumber $k_e \in [1, 2]$, $[2, 3]$, and $[3, 4]$, are separately projected to the output of the receiver. The amplitudes of simulated image transfer functions are smaller than the theoretically calculated ones mainly because of the difference between theoretical and simulated amplitudes of the filter layer transmission shown in the previous section.

In order to study the influence of varying distance D, we plot the theoretically calculated amplitude and phase (Fig. 4c) of $T_{continous}$ for lens1 ($a_1 = 4.8$ mm, $k_G = 2k_0$) when $D = 130$ and 200 mm. From Fig. 4c, we find that in continuous source case, whereas the multiple reflections between the waveguide generate small variations in the amplitude response, the major trend of band-filtering behavior does not change with distance $D$. However, changing $D$ will result in large change in the phase response, unlike that in the amplitude response. The theoretically calculated amplitudes and phases of $T_{pulse}$ for lens1 when $D = 130$ and 200 mm are also plotted in Fig. 4d. As indicated by the expression of $D_g(k_e)$, the amplitude does not change with $D$ while the phase does. From above analysis, we conclude that changing $D$ will only change phases of different wave vector components but not the amplitudes. With proper phase compensation, the distance between image and object can be changed flexibly without influencing the imaging quality.

**Subwavelength imaging and edge detection**. The designed far-field subwavelength imaging system has the ability to separate spatial frequency bands and project them to the far-field. Different spatial frequency bands represent portions of the image with different spatial resolutions. By manipulating those separated bands, various functions can be realized. For example, if we add all the separated bands together, a full image of the object can be reconstructed with subwavelength details. If only the higher spatial frequency bands are added, edge detection of the objects can be achieved, which is an important function in image processing[28–32].

We use Autodesk Ember 3D Printer to fabricate the three sets of transmitting and receiving pairs lens1, lens2 and lens3 that work for $k_e \in [1, 2]$, $[2, 3]$, and $[3, 4]$, respectively (Fig. 5). Each set is put into a waveguide of height $h = 5$ mm and width 250 mm to confine the converted propagating wave from the transmitter to the receiver. The propagating distance is $D = 130$ mm. We first use the fabricated system to image a 1D slit of size $d = 60$ mm ($1.57\lambda_0$). Four measurements are performed, three of which are from the three devices (lens1, lens2, lens3) corresponding to $k_e \in [1, 2]$, $[2, 3]$, and $[3, 4]$. Another measurement is performed when

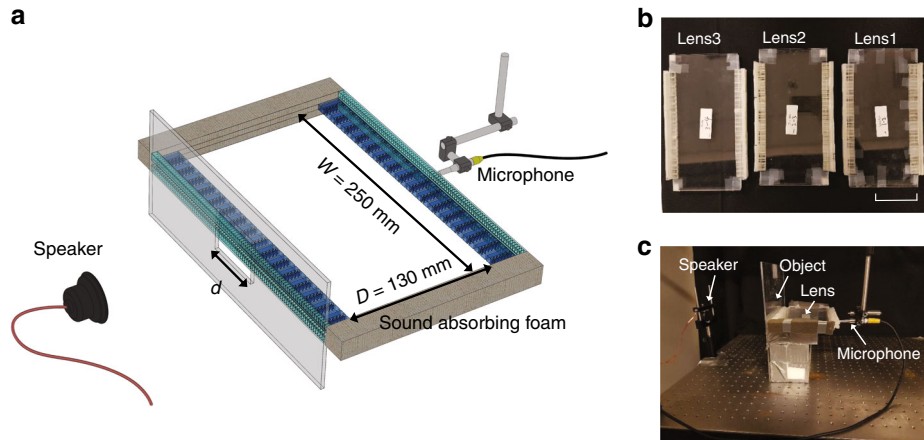

**Fig. 5** Experimental setup. **a** Experimental setup for subwavelength imaging and edge detection of 1-D slits. The speaker is 20 cm away from the lenses. The objects are 1-D slits put at the entrance of the transmitter. A microphone is used to scan the signal distribution at the exit of the receiver. **b** 3D printed lens1, lens2 and lens3. The white scale bar on the right bottom represents the length of 100 mm. **c** A photo of the experimental system

the slit is put at the entrance of the same waveguide as the other measurements but without any of the HR arrays and gratings (len0), correspoinding to the propagating components $k_e \in [0, 1]$. Each measurement gets the sound field distribution along the exit of the corresponding lens. Detailed measurement and data processing procedure are in the Methods section. Normalized spatial spectrums of the measured field distributions, $S_n'(k_e)$, $n = 0, 1, 2, 3$, are plotted in Figs. 6a–d as black solid lines. The measured spectrum of each lens verifies the spatial filtering capability of our designed system. From Figs. 6a–d we also observe that all the other higher order diffraction components are at least 15 dB smaller than those in the 1st order diffraction after passing through the filter layer at the exit end.

The normalized space domain signals $I_n^p(x)$, $n = 0, 1, 2, 3$ are obtained by taking inverse Fourier transform of the windowed spatial spectrums (See Methods) and plotted in Figs. 6e–h. By adding all four normalized signals together, we get $I_{\mathrm{full}}(x) = \sum_{n=0}^{3} I_n^p(x)$, which is a full image of the 60 mm slit, as shown in Fig. 6j. If we remove the diffraction-limited component $I_0^p$, $I_{\mathrm{edge}}(x) = \sum_{n=1}^{3} I_n^p(x)$ gives two edges of the slit having distance of 60 mm, as shown in Fig. 6i. Each edge is indicated by a peak having half-width of around $\lambda_0/4$. The signal-to-noise ratio (SNR) is defined as the ratio of peak height at the edge to the second largest peak height within $[-\lambda_0, \lambda_0]$ distance of the edge (neglecting the peak height of the other edge in this range, if exists). SNR for the 60 mm slit edge detection is calculated as 4.4 dB for the left edge and 9.5 dB for the right edge. The black dashed lines in Fig. 6 are COMSOL simulation results of spectrums and images corresponding to the experimental measurements.

We repeat the measurement process to obtain the $I_1^p$, $I_2^p$, and $I_3^p$ of slits with sizes 30 mm ($0.8\lambda_0$), 15 mm ($0.4\lambda_0$), and 10 mm ($0.26\lambda_0$), and plotted the $I_{\mathrm{edge}}(x) = \sum_{n=1}^{3} I_n^p(x)$ for the corresponding three slits in Figs. 7a–c. The edges of 30 mm and 15 mm slits are successfully detected. The SNR of the 30 mm slit is 7.5 dB for the left edge and 8.1 dB for the right edge. The SNR of the 15 mm slit is 7.5 dB for the left edge and 5.7 dB for the right edge. However the system fails to detect the edges of the 10 mm slit, since the highest spatial frequency we can detect is less than $4k_0$, corresponding to feature size larger than $\lambda_0/4 = 9.5$ mm. 10 mm is almost the limitation of detection in the current system. In Fig. 7i, we only observe one peak of size around 10 mm, instead of two edges with 10 mm distance. We then use the four lenses to perform imaging for double-slit objects. One object is composed

of one 20 mm ($0.52\lambda_0$) slit and one 30 mm ($0.8\lambda_0$) slit with 20 mm edge-to-edge distance. Another object is composed of two 10 mm ($0.26\lambda_0$) slits that have 10 mm edge-to-edge distance. We plot the experimental full images and edge images for the objects in Figs. 7d–g. The four lenses successfully capture both the edges and the full image for the 20–30 mm double slits. And as expected, our device gives the subwavelength image of the two slits but does not capture the edges due to the upper limit of wave vector range, which is $4k_0$. One can notice that the obtained images are not perfect square functions as the object, and the edges have widths of around $\lambda_0/4$ instead of being perfect pulses. Those imperfections are due to the fact that we only use four lenses to obtain spatial information up to $4k_0$. In order to fully recover the square images similar to the objects, more than four lenses that can recover deeper subwavelength information are necessary. The smaller or more complex the object is, the more spatial wave vector bands are needed in order to resolve the image.

In order to better evaluate the performance of the four lenses, we plot red dashed lines in Figs. 6e–j and 7 as theoretical references. Those red dashed lines are obtained with the corresponding $[0, k_0]$, $[k_0, 2k_0]$, $[2k_0, 3k_0]$ or $[3k_0, 4k_0]$ bands directly cut from the spatial Fourier transforms of the slits. For example, the red dashed line in Fig. 6f is obtained with the following steps: 1) Perform spatial Fourier transform for the 60 mm slits (a rectangular function with width 60 mm) to obtain the spectrum. 2) Use a window function to obtain the $[k_0, 2k_0]$ band from the spectrum. 3) Perform reverse Fourier transform for the windowed $[k_0, 2k_0]$ band to get the red dashed line in Fig. 6f showing the spatial field distribution of the $[k_0, 2k_0]$ band. We can see that the experimental outputs from the lenses are very similar to those theoretical references. The slight differences are due to the non-uniform transmission amplitudes for different wave vector components passing through the lenses. The non-uniform amplitudes are caused by non-uniform frequency response of the system for different wave vectors as well as different influence of damping on different wave vectors. The blue solid lines in Fig. 7 represent the diffraction-limited images from the empty wave-guide (lens0), from which we can verify the resolution improvement by adding the signals from lens1, lens2 and lens3.

## Discussion
The experimental results have successfully demonstrated that our designed system can separate and project different

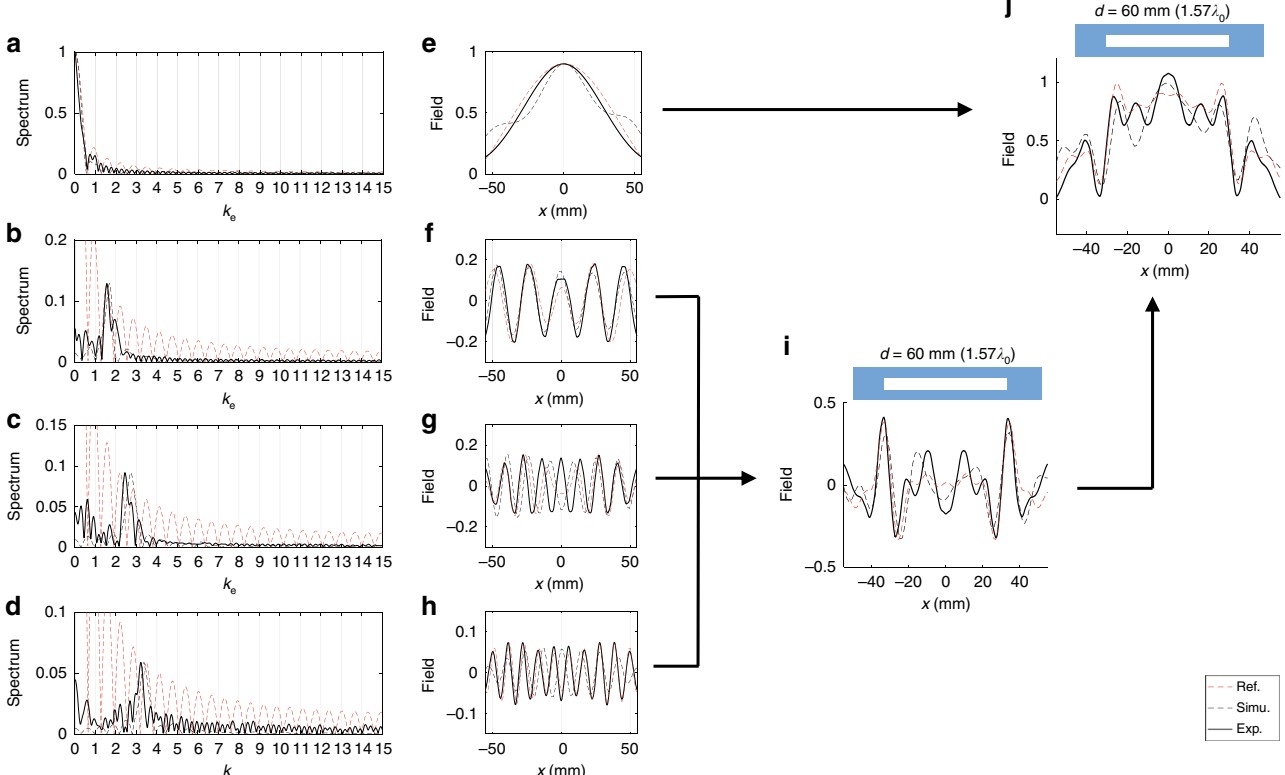

**Fig. 6** Subwavelength imaging and edge detection results of a 1-D 60 mm ($1.57\lambda_0$) slit. **a–d** When the object is a 60 mm slit, the normalized spatial spectrums of received signals from empty waveguide (**a**), lens1 (**b**), lens2 (**c**), and lens3 (**d**), respectively. **e, f** The reverse Fourier transforms of the shaded regions $k_e \in [0, 1]$ (**e**), $k_e \in [1, 2]$ (**f**), $k_e \in [2, 3]$ (**g**), and $k_e \in [3, 4]$ (**h**) in the spectrum plotted in (**a–d**) correspondingly. (i) The edge image of the 60 mm slit by adding three subwavelength components. **j** The full image of the 60 mm slit by adding subwavelength as well as propagating components. In (**a–j**), black solid lines are experimental measurements, black dashed lines are simulation results, and red dashed lines are theoretical references

subwavelength spatial frequency bands to the far-field. The distance between image and object is largely extended compared to other subwavelength imaging systems based on negative index materials, Fabry-Perot resonances, or trapped modes. The distance can now be many wavelengthes away and also be flexible according to application. Furthermore, the waveguide between the transmitter and the receiver in our system does not need to be straight as long as the height of the waveguide is smaller than the wavelength in free space. We can bend the waveguide according to the applications. The demonstrated system in this paper works in transmission mode. The system can also work in reflection mode, where only one set of filter layer and grating layer combination is needed. The spatial symmetrical part can be performed by simply adding a reflecting mirror in the far-field. Another intriguing property of our system is that the transmitter and the receiver are symmetrical. An object can be put to either side to project the subwavelength information to the other side. With this symmetry, our system can be used as long distance acoustic communication system to exchange subwavelength information. Those tunabilities in the positions of object/image and the waveguide shape give our system unprecedented flexibilities compared to other far-field imaging systems such as hyperlenses and time reversal techniques.

There are several limiting factors for the spatial resolution of the system. The first factor comes from the filter layer. As discussed earlier, the largest wave vector that can be amplified by the filter layer is limited by the periods of the filter layer, which is determined by the smallest size of Helmholtz resonators.

However, smaller Helmholtz resonator size results in higher working frequency. Higher working frequency will further result in smaller wave vector range. In order to increase the spatial resolution, resonators in the filter layer should be able to generate lower resonance frequency with smaller size. The second factor comes from the grating layer. The largest wave vector that can be converted back to propagating wave is determined by the grating period, which is limited by the smallest lateral size of each phase delay channel. The loss is another factor that will influence the resolution since higher spatial frequency components will be attenuated more by the thermal loss in each component.

In summary, we have designed and tested an acoustic subwavelength imaging system based on wave vector conversion. The combination of resonator arrays that enhance the waves with specific wave vectors, and binary phase gratings that add or subtract wave vectors by first order diffraction, establishes a one-to-one relationship between the subwavelength and propagating wave components. All theoretical, simulation and experimental data confirmed the capability our system in separating and projecting different subwavelength wave vector bands to the far-field of the object. We have demonstrated the application of our system with the edge detection of acoustic scattering objects with resolution upper limit of $\lambda_0/4$ and signal-to-noise ratio of ~6 dB. The system can be scaled to sizes of interest in ultrasonic medical imaging and non-destructive testing applications to increase the imaging resolution, and be applied to acoustic communication systems to increase the information capacity by incorporating subwavelength information.

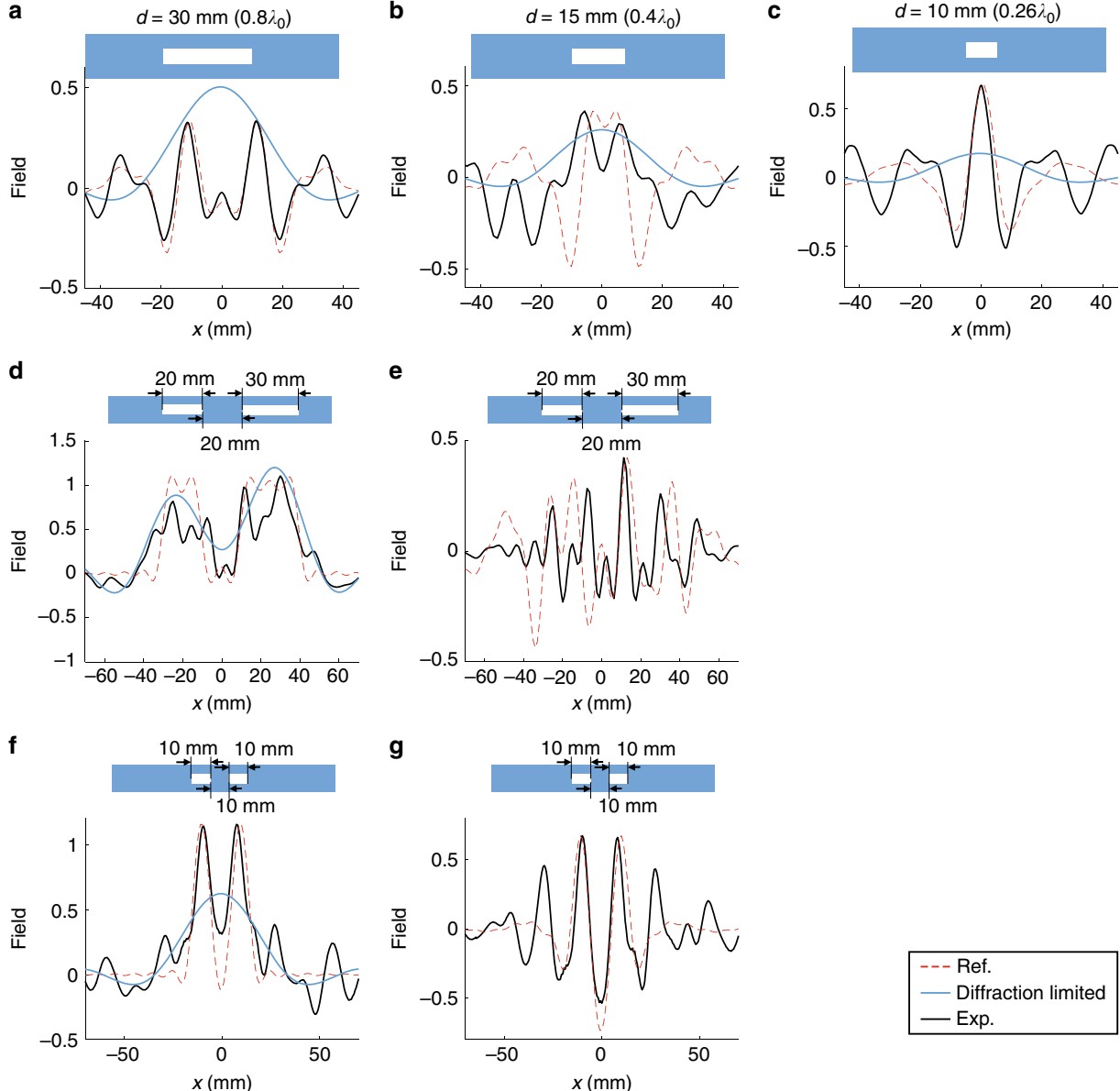

**Fig. 7** Subwavelength imaging and edge detection for 1-D single slits and double slits. **a–c** Edge images for a single slit of size 30 mm (**a**), 15 mm (**b**), and 10 mm (**c**), respectively. **d** The full image of a double-slit object with 20 mm and 30 mm slit widths and 20 mm edge-to-edge distance. **e** The edge image of the 20–30 mm double slits. **f** The full image of a double-slit object with 10 mm and 10 mm slit widths and 10 mm edge-to-edge distance. **g** The edge image of the 10–10 mm double slits. In (**a–g**), black solid lines are experimental measurements, blue solid lines are diffraction-limited results from empty waveguide, and red dashed lines are theoretical references

## Methods

**Experimental setup**. As shown in Fig. 5a, the slit to be imaged is put in contact with the HRs in the transmitter (len1, lens2 and lens3) or the entrance of the empty waveguide (lens0). The left and right sides of the waveguide are covered by sound absorbing foam. A sound speaker located 20 cm (around five wavelengths) away from the slit sends eight periods of 9000 Hz sound wave. The wave incidents on the slit and the scattered wave from the slit enters the transmitter. At the receiver, a microphone (PCB 130F20) scans the sound field along the $x$ direction at the exit of the HR arrays. The scanning has resolution of 1.25 mm and in total 180 points (22.5 cm) are scanned. At each scanning point, 0.01 s (90 periods for wave of frequency 9000 Hz) of the sound wave is recorded.

**Experimental data normalization**. A time window of 10 periods is applied to each directly measured signal to obtain the portion of sound signal that comes from the imaging system in order to eliminate the interference of reflected wave from surrounding environment (the lab walls, table surface, etc.) and the multiple reflections inside the waveguides. By taking the time domain Fourier transform of the time-windowed signals, the complex values of the 9000 Hz components at all

scanning points are obtained as $I_n(x)$, where $n = 0, 1, 2, 3$ (device number). Spatial Fourier transforms of $I_n(x)$ are obtained as $S_n(k_e)$, $n = 0, 1, 2, 3$. The 1-D spatial Fourier transform of a rectangular function of 60 mm width, $S_r(k_e)$, is used as reference for normalization (plotted as red dashed lines in Figs. 6a–d). We normalize the amplitude of the measured spectrum $S_n(k_e)$ to the amplitude of the reference $S_r(k_e)$, and compensate for the phase variations of different wave vectors caused by passing through the system. Each $S_n(k_e)$ is divided by the factor $A_n = \frac{\max|S_n(k_e)|}{\max|S_r(k_e)|} e^{j\phi_n(k_e)}$, where $\frac{\max|S_n(k_e)|}{\max|S_r(k_e)|}$ is the ratio of the maximum amplitudes of $S_n(k_e)$ and $S_r(k_e)$ in the range $k_e \in [n-1, n+1]$, $n = 1, 2, 3$. The time windowed signal has 10 periods, which is shorter than the time needed for second arrival from waveguide internal reflection ($3D/c_{air}$). So $\phi_n(k_e)$ is the phase of image transfer function $T_{pulse} = T_f D_g T_f$ theoretically calculated for lens1, lens2, and lens3. When $n = 0$, $\phi_0(k_e) = e^{-j(D+2L+2a_1)\sqrt{k_0^2-(k_e k_0)^2}}$ ($k_e \in [-1, 1]$) is the phase profile after wave propagates through distant $D + 2L + 2a_1$ in an empty waveguide, where $L$ is the length of the grating layer in $z$ direction and $a_1$ is the size of the filter layer in $z$ direction. The $A_n$ calculated here using the slit of size $d = 60$ mm are stored and used as normalization factors for imaging processes of other slit sizes. The normalized spectrums are denoted by $S'_n(k_e)$, $n = 0, 1, 2, 3$. Multiplying $S'_n(k_e)$ with a rectangular function $R_n(k_e)$ ($R_n(k_e) = 1$ when $|k_e| \in [n-1, n]$ and $R_n(k_e) = 0$

elsewhere) to eliminate the components outside the range $|k_e| \in [n-1, n]$, and performing the inverse spatial Fourier transform for $S'_n(k_e)R_n(k_e)$, we obtain the processed space domain signal $I^p_n(x)$, $n = 0, 1, 2, 3$.

## Data Availability
The data that support the findings of this study are available from the corresponding authors upon reasonable request.

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

## Acknowledgements
C.M., S.K. and N.X.F. acknowledge the Multidisciplinary University Research Initiative from the Office of Naval Research for financial support through Grant No. N00014-13-1-0631.

## Author contribution
C.M. and N.X.F conceived the idea and designed the research. C.M. performed the theoretical study and numerical simulation. S.K. and C.M. fabricated the lenses for experiment. C.M. carried out the experiment. C.M. and N.X.F. drafted the manuscript and all authors contributed to the writing of the manuscript.

## Additional information

**Competing interests:** The authors declare no competing interests.

