## [Peer Review File · Nature Communications]

Reviewers' comments:

Reviewer #1 (Remarks to the Author):

The authors proposed and realized a subwavelength edge detection imaging system in which the subwavelength information from the object is converted to far-field through a filter layer and a grating layer and the reverse conversion is done at the far-end of the system.

The proof-of-concept experiment is novel in the way that we do not need additional complicated algorithm in getting the final image, which can now directly revealed by just adding the several images together. The limitation in resolution can also be improved later by having finer structures. The results are convincing in edge detection and I believe the work will inspire more developments along this direction in both acoustics and optics. So, I strongly support its publication but at the same time I would like the authors to clarify the following questions on the working principle and also its application.

The grating layer converts the high-k component to propagating waves at the entrance end. I can imagine that other evanescent waves components decay in the far-field propagation. How about the conversion of the grating layer at the exit end? The propagating wave in this case is converted to evanescent waves of target k. Will there be other undesired evanescent wave components from this process and how to evaluate these?

Can the authors also comment on whether the system can be used for resolving two narrow slits with subwavelength resolution if we add back the original far-field PO information?

Reviewer #2 (Remarks to the Author):

The paper presents a subwavelength imaging technique based on spatial filtering and wave vector conversion. The spatial filtering is performed using an array of Helmholtz resonators and the evanescent-to-propagative wave vector conversion is performed by a binary phase grating. The proposed device is able to separate different spatial frequencies and project them to the far-field. In addition, the authors show the possibility to perform edge detection by removing the fundamental wave vector component.

In my opinion, the underlying physics is new and interesting, and the proposed device offers unique imaging capabilities compared to existing subwavelength imaging techniques. Moreover, the theoretical, numerical and experimental results are in good agreement and convincingly support the major claims of the paper. Therefore, I support publication of the manuscript after addressing some minor points:

1: Could the authors explain (or give a reference that explains) how the expression for P_0 , P_2 , etc are derived?

2: The authors claim that the proposed device is able to perform far-field subwavelength imaging. This is due to the fact that the receiver/transmitter pair can be separated by a distance much larger than the operating wavelength. However, the transmitter must be in contact with the imaged object (i.e. in its near-field). Therefore, can one still consider the technique as a far-field one?

3. Even if the experimental results in Fig. 6 clearly demonstrate the subwavelength imaging and edge detection capabilities of the device, the experimental acoustic fields are considerably different from the theoretical ones. Could the authors add a comment on possible reasons for this

discrepancy?

Reviewer #3 (Remarks to the Author):

In this paper, the authors present a technique for imaging subwavelength objects from the far field. The idea is similar to the farfield superlens that uses gratings to convert nearfield components to propagating ones but with some tweaks. They choose to experiment with the idea in acoustic wave. A filter layer is used, with the purpose to select only a narrow range of wave vector from outside the "light cone". The wave vector is then converted to far field with binary phase grating for imaging in the far field. I think in principle this idea should work. And the authors put extensive works in theories and to characterize the system with numerical simulations. The paper is also mostly nicely crafted, with each component clearly introduced then integrated together. However, the measured results in Fig 6 just don't justify that subwavelength imaging is indeed obtained. And this is the crucial factor that prevents me from recommending its publication.

First of all, coordinates are missing for Fig 6(e)-(i). Second, the slit is really the simplest kind of object to be imaged. Yet none of the results can be regarded as an image of the slit. There are basically rapid oscillations. From this one can probably guess that something is happening at the location with fast change in the field pattern, but without any prior knowledge, it's really difficult to tell with certainty what is going on. (And that is the purpose of imaging.) Also, to see the effectiveness of the idea, I suggest showing the results of individual lenses, then see what they add up to. And they should be compared with the image without the lens. A more complex object (say, at least two slits with different size and with subwavelength separation) is ideal to show the effectiveness of the lenses.

I am guessing that the undesirable experimental outcome is caused by the large overlap of transmission of the gratings. In Fig. 3b-e, one can see that the plateaus for different gratings are not separated in k-domain, but overlapped together. Perhaps this overlap causes some wavevectors to be added multiple times to form the image in Fig. 6.

I also do not understand Eq (1) on Page 7. When the denominator goes to zero, the transmission is obviously infinitely large. This cannot be true. Please explain and revise.

In short, if the experiment can be considerably improved, I will be happy to recommend the publication of this paper.

Response to Referee #1's comments:

The authors thank the reviewer's support for publication and constructive comments. We have clarified all the points and questions raised by the reviewer and reflected them in the revised manuscript. Modifications made in the revision are highlighted in blue. We thank the reviewer again for taking time to carefully review our manuscript. We hope that our response has addressed all the previous concerns.

Comment 1: The grating layer converts the high-k component to propagating waves at the entrance end. I can imagine that other evanescent waves components decay in the far-field propagation. How about the conversion of the grating layer at the exit end? The propagating wave in this case is converted to evanescent waves of target k. Will there be other undesired evanescent wave components from this process and how to evaluate these?

Response: We want to emphasize that there are two layers (filter layer and grating layer) working collectively. While the grating layer is responsible for converting different diffraction orders, the design of the filter layer only allows a specific pass band of the wave numbers that locates in the first order diffraction. At the exit end, undesired evanescent wave components from the grating layer are attenuated after passing through the filter layer. We have added more explanation for the collective working principle of the filter layer and the grating layer in the revised manuscript.

In order to evaluate the attenuation of higher order diffractions, we extended the wavenumber range in the spatial spectrum plot in Fig. 6(a-d) to $k_e = 15$. From Fig. 6(a-d) we can clearly see that only those wave components of the 1st order diffraction are detected in the received signal. All the other higher order diffraction components are at least 15dB smaller than the 1st order diffraction after passing through the filter layer at the exit end.

After revision:

On page 5, line 108-112:

The amplitudes $p_1(k)$, $p_2(k')$ and $p_3(k)$ are derived in later sections as transmission coefficients of filter layer and grating layer. The grating will also generate other higher odd order diffractions (3th order, 5th order,...). Then the filter layer in the receiver will perform post-filtering for the waves with wave number $|k| \in [k_G - k_0, k_G]$ or $[k_G, k_G + k_0]$ in the first order diffraction. Other higher order diffractions are all attenuated after passing through the filter layer at the receiver. In order to prevent the overlapping....

On page 15, line 303-305:

The measured spectrum of each lens verifies the spatial filtering capability of our designed system. From Fig. 6a-6d we also observe that all the other higher order diffraction components are at least 15dB smaller than those in the 1st order diffraction after passing through the filter layer at the exit end.

On page 17, Fig. 6(a-d):

Comment 2: Can the authors also comment on whether the system can be used for resolving two narrow slits with subwavelength resolution if we add back the original far-field P0 information?

Response: In order to answer this question, we have performed additional two sets of imaging experiments for double-slit objects. One object is composed of one 20mm slit ($0.52\lambda_0$) and one 30mm slit ($0.8\lambda_0$) with 20mm edge-to-edge distance. Another object is composed of two 10mm slits ($0.26\lambda_0$) with 10mm edge-to-edge distance. We plot the experimental full images and edge

images for the objects in Fig. 7(d-g). The full images are obtained by adding the output from all four lenses and the edge images are obtained by removing the propagating component from empty waveguide (lens0). The four lenses successfully capture both the edges and the full image for the 20mm-30mm double slits. And as expected, our device gives the subwavelength image of the 10mm-10mm slits but does not capture the edges due to the upper limit of wave vector range, which is $4k_0$.

After revision:

On page 15, line 326-333:

..., instead of two edges with 10mm distance. We then use the four lenses to perform imaging for double-slit objects. One object is composed of one 20mm slit ($0.52\lambda_0$) and one 30mm slit ($0.8\lambda_0$) with 20mm edge-to-edge distance. Another object is composed of two 10mm slits ($0.26\lambda_0$) with 10mm edge-to-edge distance. We plot the experimental full images and edge images for the objects in Fig. 7d-7g. The four lenses successfully capture both the edges and the full image for the 20mm-30mm double slits. And as expected, our device gives the subwavelength image of the 10mm-10mm slits but does not capture the edges due to the upper limit of wave vector range, which is $4k_0$.

On page 18, Fig. 7(d-g):

Response to Referee #2's comments:

The authors thank the reviewer's support for publication and constructive comments. We have clarified all the points and questions raised by the reviewer and reflected them in the revised manuscript. Modifications made in the revision are highlighted in blue. We thank the reviewer again for taking time to carefully review our manuscript. We hope that our response has addressed all the previous concerns.

Comment 1: Could the authors explain (or give a reference that explains) how the expression for P_0 , P_2 , etc are derived?

Response: The expressions $P_0 = \int_{-\infty}^{\infty} p_0(k) e^{j(kx-2\pi f_0 t)} dk$, $P_1 = \int_k p_1(k) e^{j(kx-2\pi f_0 t)} dk$,

$P_2 = \int_k p_2(k') e^{j(k'x-2\pi f_0 t)} dk'$, $P_3 = \int_k p_3(k) e^{j(kx-2\pi f_0 t)} dk$ are the integrations of Fourier spectrums, based on the principle of spatial Fourier transform [Goodman, Joseph. "Introduction to Fourier optics." (2008)]. The transition from k to k' indicates the wave vector conversion after the wave passes through binary phase gratings. The amplitudes $p_0(k)$, $p_1(k)$, $p_2(k')$ and $p_3(k)$ are derived in later sections as transmission coefficients of filter layers and grating layers. We have added sentences to the revised manuscript in order to clarify the reviewer's question.

After revision:

Page 5, line 91-92:

The scattered sound wave from the object, $P_0 = \int_{-\infty}^{\infty} p_0(k) e^{j(kx-2\pi f_0 t)} dk$, enters the filter layer in the transmitter. P_0 is expressed as the integration of the Fourier spectrum for the scattered wave from the object, based on the principle of spatial Fourier transform [24]. P_0 contains...

Page 5, line 106-108:

The amplitudes $p_1(k)$, $p_2(k')$ and $p_3(k)$ are derived in later sections as transmission coefficients of filter layers and grating layers.

Comment 2: The authors claim that the proposed device is able to perform far-field subwavelength imaging. This is due to the fact that the receiver/transmitter pair can be separated by a distance much larger than the operating wavelength. However, the transmitter must be in contact with the imaged object (i.e. in its near-field). Therefore, can one still consider the technique as a far-field one?

Response: We thank the reviewer for pointing out the limitation of the current technique. The transmitter in the near field of the object is necessary to collect and convert the evanescent waves before they are attenuated. However, the technique proposed in our work allows the receiver to be many wavelengths away from the object and the location can be flexible. Our scenario is similar to the oil-immersion or solid-immersion objectives for light microscope where the sample is in direct contact with the imaging element. It is still reasonable to call it far-field lens. Our technique will provide more flexibility compared to near-field scanning techniques such as near-

field scanning microscopes [Günther, P., U. Ch Fischer, and K. Dransfeld. "Scanning near-field acoustic microscopy." *Applied Physics B* 48.1 (1989): 89-92.] and superlenses [Zhang, Xiang, and Zhaowei Liu. "Superlenses to overcome the diffraction limit." *Nature materials* 7.6 (2008): 435.].

After revision:

On page 3, line 76-79:

In the proposed system, wave propagation will take place in empty waveguide with much less thermal viscous loss than in metamaterials. We will show that while the transmitter still needs to be close to the object, the receiver can be many wavelength away from the object and the distance can be flexible. This is the reason why we call the designed system far-field lenses.

Comment 3: Even if the experimental results in Fig. 6 clearly demonstrate the subwavelength imaging and edge detection capabilities of the device, the experimental acoustic fields are considerably different from the theoretical ones. Could the authors add a comment on possible reasons for this discrepancy?

Response: The authors thank the reviewer for pointing out the discrepancy. In our proof of concept experiments, we tested the imaging capability with only four lenses that can recover the spatial information in the range of $[0, 4k_0]$. In order to fully recover the square images similar to the objects (the "theoretical ones" mentioned by the reviewer), more than four lenses that can capture deeper subwavelength information are necessary. The smaller or more complex the object is, the more spatial wave vector bands are needed in order to resolve the image. This is the main reason for the discrepancy between the image and the object. Another reason for the discrepancy is that different wave vector components will have different transmission amplitudes after passing through the lenses. The difference in amplitudes is caused by non-uniform frequency response of the system for different wave vectors as well as different influence of damping on different wave vectors. We have added those possible reasons to the revised manuscript.

After revision:

On page 16, line 333-339:

One can notice that the obtained images are not perfect square functions as the object, and the edges have widths of around $\lambda_0/4$ instead of perfect pulses. Those imperfections are due to the fact that we only use four lenses to obtain spatial information up to $4k_0$. In order to fully recover the square images similar to the objects, more than four lenses that can recover deeper subwavelength information are necessary. The smaller or more complex the object is, the more spatial wave vector bands are needed in order to resolve the image.

On page 16, line 340-352:

In order to better evaluate the performance of the four lenses, we plot red dashed lines in Fig. 6e-6j and Fig. 7 as theoretical references. Those red dashed lines are obtained with the

corresponding $[0, k_0]$, $[k_0, 2k_0]$, $[2k_0, 3k_0]$, $[3k_0, 4k_0]$ bands directly cut from the spatial Fourier transforms of the slits. For example, the red dashed line in Fig. 6(f) is obtained with the following steps: 1) Perform spatial Fourier transform for the 60mm slits to obtain the spectrum. 2) Use a window function to obtain the $[k_0, 2k_0]$ band from the spectrum. 3) Perform reverse Fourier transform for the windowed $[k_0, 2k_0]$ band to get the red dashed line in Fig. 6(f) showing the spatial field distribution of the $[k_0, 2k_0]$ band. We can see that the experimental outputs from the lenses are very similar to those theoretical references. The slight differences are due to the non-uniform transmission amplitudes for different wave vector components passing through the lenses. The non-uniform amplitudes are caused by non-uniform frequency response of the system for different wave vectors as well as different influence of damping on different wave vectors.

Response to Referee #3's comments:

The authors thank the reviewer's kind and constructive comments. We have clarified all the points and questions raised by the reviewer and reflected them in the revised manuscript. Modifications made in the revision are highlighted in blue. We thank the reviewer again for taking time to carefully review our manuscript. We hope that our response has addressed all the previous concerns.

Comment 1: First of all, coordinates are missing for Fig 6(e)-(i)

Response: We have added coordinates to Fig. 6 and Fig. 7 as shown in the revised manuscript.

Comment 2: Second, the slit is really the simplest kind of object to be imaged. Yet none of the results can be regarded as an image of the slit. There are basically rapid oscillations. From this one can probably guess that something is happening at the location with fast change in the field pattern, but without any prior knowledge, it's really difficult to tell with certainty what is going on. (And that is the purpose of imaging.)

Response: We thank the reviewer for pointing out the non-perfect performance of the current system.

Regarding to reviewer #3's comment that our designed devices fail to perform imaging for the slits if we have no prior knowledge, it is our understanding that common imaging processes are all based on some prior knowledge, such as the range of dimensions of the object, the surrounding environment, etc. In biomedical imaging, we will know in prior the types of tissue we are trying to image, the acoustic impedance contrast, even the typical shapes (round, elliptical, etc). Imaging devices are designed to increase the detected information based on the prior knowledge we have of the object. For example, in our cases, the prior knowledge is that the object is 1-D slits. What we are presenting is a resolution enhancement technique, whereas a solution to identify fully unknown media is beyond the scope of this work.

In the revised manuscript, we have added comments about the reasons that causing the discrepancy between the images obtained from the four lenses and the objects. The non-perfect performance of our system is due to the fact that we tested the imaging capability with only four lenses that can recover the spatial information in the range of $[0, 4k_0]$. In order to fully recover the square images of the slits, more than four lenses that can capture deeper subwavelength information are necessary. The smaller or more complex the object is, the more spatial wave vector bands are needed in order to resolve the image. While our technique can be easily extended to higher spatial resolution components by adjusting the geometric parameters of the grating layer and filter layer, we want to point out that for the purpose of edge detection, perfect images of the edges are not necessary. Peaks with subwavelength width and largest amplitudes among all the peaks that can indicate the locations of the edges are enough. As shown in our proof of concept experiments, we have located the edges of subwavelength slits with peaks of around $\lambda_0/4$.

In order to better illustrate the performance of the four lenses, we also added a set of theoretical reference lines to Fig. 6 and Fig. 7 for comparison. The red dashed lines in Fig. 6 and 7 are obtained with the corresponding $[0, k_0]$, $[k_0, 2k_0]$, $[2k_0, 3k_0]$, $[3k_0, 4k_0]$ bands directly cut from the spatial Fourier transforms of the slits. For example, the red dashed line in Fig. 6(f) is obtained with the following steps:

- 1) Perform spatial Fourier transform for the 60mm slits to obtain the spectrum.
- 2) Use a window function to obtain the $[k_0, 2k_0]$ band from the spectrum.
- 3) Perform reverse Fourier transform for the $[k_0, 2k_0]$ band to get the red dashed line in Fig. 6(f) showing the spatial field distribution.

We can see that the outputs from the lenses are very similar to those theoretical references corresponding to same wave vector bands. The slight difference are due to the different transmission amplitudes for different wave vector components passing through the lenses.

After revision:

On page 16, line 333-339:

One can notice that the obtained images are not perfect square functions as the object, and the edges have widths of around $\lambda_0/4$ instead of perfect pulse. Those imperfections are due to the fact that we only use four lenses to obtain spatial information up to $4k_0$. In order to fully recover the square images similar to the objects, more than four lenses that can recover deeper subwavelength information are necessary. The smaller or more complex the object is, the more spatial wave vector bands are needed in order to resolve the image.

On page 16, line 340-352:

In order to better evaluate the performance of the four lenses, we plot red dashed lines in Fig. 6e-6j and Fig. 7 as theoretical references. Those red dashed lines are obtained with the corresponding $[0, k_0]$, $[k_0, 2k_0]$, $[2k_0, 3k_0]$, $[3k_0, 4k_0]$ bands directly cut from the spatial Fourier transform of the slits. For example, the red dashed line in Fig. 6(f) is obtained with the following steps: 1) Perform spatial Fourier transform for the 60mm slits to obtain the spectrum. 2) Use a window function to obtain the $[k_0, 2k_0]$ band from the spectrum. 3) Perform reverse Fourier transform for the $[k_0, 2k_0]$ band to get the red dashed line in Fig. 6(f) showing the spatial field distribution of the $[k_0, 2k_0]$ band. We can see that the outputs from the lenses are very similar to those theoretical references. The slight difference are due to the non-uniform transmission amplitudes for different wave vector components passing through the lenses. The non-uniform amplitudes are caused by non-uniform frequency response of the system for different wave vectors as well as different influence of damping on different wave vectors.

On page 17, Fig. 6(e-j)

On page 18, Fig. 7:

Comment 3: Also, to see the effectiveness of the idea, I suggest showing the results of individual lenses, then see what they add up to. And they should be compared with the image without the lens.

Response: Following the suggestion from the reviewer, we have plot the results of individual lenses for the slit of 60mm. The reference, simulation and experiment agree well. The slight difference are due to the different transmission amplitudes for different wave vector components passing through the lenses.

In Fig. 7, the blue solid lines represent the diffraction limited images from an empty waveguide. We can clearly observe the resolution improvement by adding the signals from lens1, lens2 and lens3.

After revision:

On page 15, line 307-308:

The normalized space domain signals $I_n^p(x)$, $n = 0,1,2,3$ are obtained by taking inverse Fourier transform of the windowed spatial spectrums (See Methods) and plotted in Fig. 6e-6h.

On page 16, line 352-355:

The blue solid lines in Fig. 7 represent the diffraction-limited images from the empty waveguide (lens0), from which we can verify the resolution improvement by adding the signals from lens1, lens2 and lens3.

Comment 4: A more complex object (say, at least two slits with different size and with subwavelength separation) is ideal to show the effectiveness of the lenses.

Response: We thank reviewer for the constructive suggestion for showing the effectiveness of the lenses. Following the suggestion of the reviewer, we have performed additional two sets of imaging experiments for two double-slit objects. One object is composed of one 20mm slit ($0.52\lambda_0$) and one 30mm slit ($0.8\lambda_0$) with 20mm edge-to-edge distance. Another object is composed of two 10mm slits ($0.26\lambda_0$) with 10mm edge-to-edge distance. We plot the experimental full images and edge images for the objects in Fig. 7(d-g). The full images are obtained by adding the output from all four lenses and the edge images are obtained by removing the propagating component from lens0. The four lenses have successfully recovered both the edges and the full image for the 20mm-30mm double slits, and recovered the full image for the 10mm-10mm double slits.

After revision:

On page 15, line 326-333:

..., instead of two edges with 10mm distance. We then use the four lenses to perform imaging for double-slit objects. One object is composed of one 20mm slit ($0.52\lambda_0$) and one 30mm slit ($0.8\lambda_0$) with 20mm edge-to-edge distance. Another object is composed of two 10mm slits ($0.26\lambda_0$) with 10mm edge-to-edge distance. We plot the experimental full images and edge images for the objects in Fig. 7d-7g. The four lenses successfully capture both the edges and the full image for the 20mm-30mm double slits. And as expected, our device gives the subwavelength image of the 10mm-10mm slits but does not capture the edges due to the upper limit of wave vector range, which is $4k_0$.

Comment 5: I am guessing that the undesirable experimental outcome is caused by the large overlap of transmission of the gratings. In Fig. 3b-e, one can see that the plateaus for different gratings are not separated in k-domain, but overlapped together. Perhaps this overlap causes some wave vectors to be added multiple times to form the image in Fig. 6.

Response: Fig.3b-e plot the transmission coefficients for the grating layers itself without any filter layer. The overlap of different gratings in k-domain is removed by the filter layers next to

each grating. The final image transfer functions are shown in Fig. 4a-b in the revised manuscript, where no overlap is observed.

Comment 6: I also do not understand Eq (1) on Page 7. When the denominator goes to zero, the transmission is obviously infinitely large. This cannot be true. Please explain and revise.

Response: Equation (1) is derived based on the assumptions that no damping is considered and the array is infinitely long. This is comparable to eigenmode analysis. In realistic experiments or simulations, the arrays are of finite length and damping has to be considered, so that the transmission will not diverge. The large amplitude indicates the resonating coupling to the HR array. We have clarified this question in the revised manuscript.

After revision:

On page 8, line 158-162:

When the denominator ($d_1^2 k_e^2 k_0^2 + 2YZ$) approaches zero, the resonance occurs and the transmission through the waveguide between the HR arrays is largely enhanced. Eq. 1 is derived based on the assumptions that no damping is considered and the array is infinitely long. This is comparable to eigenmode analysis. In realistic experiments or simulations, the arrays are of finite length and damping has to be considered, so that the transmission will not diverge.

REVIEWERS' COMMENTS:

Reviewer #2 (Remarks to the Author):

The authors have convincingly addressed all the points raised in my previous review, and they have corrected the manuscript accordingly. Therefore I support publication of the manuscript as is.

Reviewer #3 (Remarks to the Author):

I am glad to see that the authors have made substantial efforts to improve this work. In particular, the double-slit imaging experiment has wonderful results that convincingly show the effectiveness of the approach. I think the paper is now a nice piece and I recommend its publication.